# Joint Training of a Convolutional Network and a Graphical Model for Human Pose Estimation

**Jonathan Tompson, Arjun Jain, Yann LeCun, Christoph Bregler**
New York University
{tompson, ajain, yann, bregler}@cs.nyu.edu

## Abstract

This paper proposes a new hybrid architecture that consists of a deep Convolutional Network and a Markov Random Field. We show how this architecture is successfully applied to the challenging problem of articulated human pose estimation in monocular images. The architecture can exploit structural domain constraints such as geometric relationships between body joint locations. We show that joint training of these two model paradigms improves performance and allows us to significantly outperform existing state-of-the-art techniques.

## 1   Introduction

Despite a long history of prior work, human body pose estimation, or specifically the localization of human joints in monocular RGB images, remains a very challenging task in computer vision. Complex joint inter-dependencies, partial or full joint occlusions, variations in body shape, clothing or lighting, and unrestricted viewing angles result in a very high dimensional input space, making naive search methods intractable.

Recent approaches to this problem fall into two broad categories: 1) more traditional deformable part models [27] and 2) deep-learning based discriminative models [15, 30]. Bottom-up part-based models are a common choice for this problem since the human body naturally segments into articulated parts. Traditionally these approaches have relied on the aggregation of hand-crafted low-level features such as SIFT [18] or HoG [7], which are then input to a standard classifier or a higher level generative model. Care is taken to ensure that these engineered features are sensitive to the part that they are trying to detect and are invariant to numerous deformations in the input space (such as variations in lighting). On the other hand, discriminative deep-learning approaches learn an empirical set of low and high-level features which are typically more tolerant to variations in the training set and have recently outperformed part-based models [27]. However, incorporating priors about the structure of the human body (such as our prior knowledge about joint inter-connectivity) into such networks is difficult since the low-level mechanics of these networks is often hard to interpret.

In this work we attempt to combine a *Convolutional Network* (ConvNet) Part-Detector – which alone outperforms all other existing methods – with a part-based Spatial-Model into a unified learning framework. Our translation-invariant ConvNet architecture utilizes a multi-resolution feature representation with overlapping receptive fields. Additionally, our Spatial-Model is able to approximate MRF loopy belief propagation, which is subsequently back-propagated through, and learned using the same learning framework as the Part-Detector. We show that the combination and joint training of these two models improves performance, and allows us to significantly outperform existing state-of-the-art models on the task of human body pose recognition.

## 2   Related Work

For unconstrained image domains, many architectures have been proposed, including "shape-context" edge-based histograms from the human body [20] or just silhouette features [13]. Many techniques have been proposed that extract, learn, or reason over entire body features. Some use a combination of local detectors and structural reasoning [25] for coarse tracking and [5] for person-dependent tracking). In a similar spirit, more general techniques using "Pictorial Structures" such as the work by Felzenszwalb et al. [10] made this approach tractable with so called 'Deformable Part Models (DPM)'. Subsequently a large number of related models were developed [1, 9, 31, 8]. Algorithms which model more complex joint relationships, such as Yang and Ramanan [31], use a flexible mixture of templates modeled by linear SVMs. Johnson and Everingham [16] employ a cascade of body part detectors to obtain more discriminative templates. Most recent approaches aim to model higher-order part relationships. Pishchulin [23, 24] proposes a model that augments the DPM model with *Poselet* [3] priors. Sapp and Taskar [27] propose a multi-modal model which includes both holistic and local cues for mode selection and pose estimation. Following the *Poselets* approach, the *Armlets* approach by Gkioxari et al. [12] employs a semi-global classifier for part configuration, and shows good performance on real-world data, however, it is tested only on arms. Furthermore, all these approaches suffer from the fact that they use hand crafted features such as HoG features, edges, contours, and color histograms.

The best performing algorithms today for many vision tasks, and human pose estimation in particular ([30, 15, 29]) are based on deep convolutional networks. Toshev et al. [30] show state-of-art performance on the 'FLIC' [27] and 'LSP' [17] datasets. However, their method suffers from inaccuracy in the high-precision region, which we attribute to inefficient direct regression of pose vectors from images, which is a highly non-linear and difficult to learn mapping.

Joint training of neural-networks and graphical models has been previously reported by Ning et al. [22] for image segmentation, and by various groups in speech and language modeling [4, 21]. To our knowledge no such model has been successfully used for the problem of detecting and localizing body part positions of humans in images. Recently, Ross et al. [26] use a message-passing inspired procedure for structured prediction on computer vision tasks, such as 3D point cloud classification and 3D surface estimation from single images. In contrast to this work, we formulate our message-parsing inspired network in a way that is more amenable to back-propagation and so can be implemented in existing neural networks. Heitz et al. [14] train a cascade of off-the-shelf classifiers for simultaneously performing object detection, region labeling, and geometric reasoning. However, because of the forward nature of the cascade, a later classifier is unable to encourage earlier ones to focus its effort on fixing certain error modes, or allow the earlier classifiers to ignore mistakes that can be undone by classifiers further in the cascade. Bergtholdt et al. [2] propose an approach for object class detection using a parts-based model where they are able to create a fully connected graph on parts and perform MAP-inference using $A^*$ search, but rely on SIFT and color features to create the unary and pairwise potentials.

## 3   Model

### 3.1   Convolutional Network Part-Detector

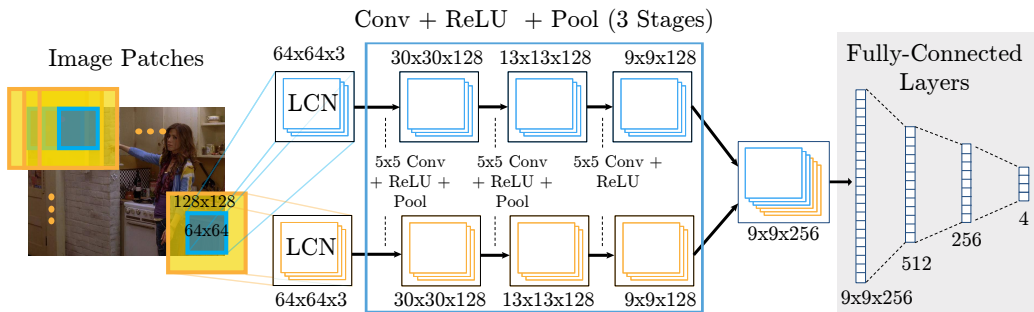

Figure 1: Multi-Resolution Sliding-Window With Overlapping Receptive Fields

The first stage of our detection pipeline is a deep ConvNet architecture for body part localization. The input is an RGB image containing one or more people and the output is a *heat-map*, which produces a per-pixel likelihood for key joint locations on the human skeleton.

A *sliding-window* ConvNet architecture is shown in Fig 1. The network is slid over the input image to produce a dense heat-map output for each body-joint. Our model incorporates a *multi-resolution* input with *overlapping receptive fields*. The upper convolution bank in Fig 1 sees a standard 64x64 resolution input window, while the lower bank sees a larger 128x128 input context down-sampled to 64x64. The input images are then Local Contrast Normalized (LCN [6]) (after down-sampling with anti-aliasing in the lower resolution bank) to produce an approximate Laplacian pyramid. The advantage of using overlapping contexts is that it allows the network to see a larger portion of the input image with only a moderate increase in the number of weights. The role of the Laplacian Pyramid is to provide each bank with non-overlapping spectral content which minimizes network redundancy.

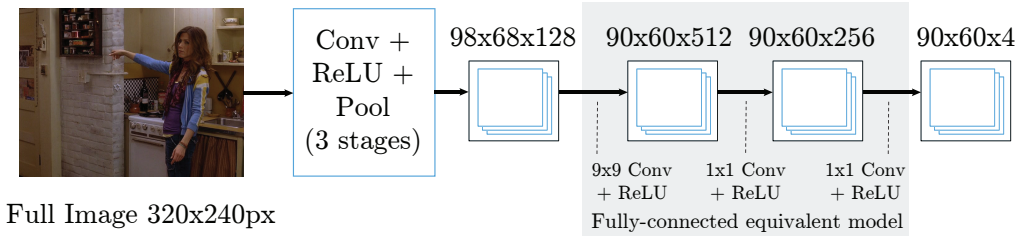

Figure 2: Efficient Sliding Window Model with Single Receptive Field

An advantage of the Sliding-Window model (Fig 1) is that the detector is translation invariant. However a major drawback is that evaluation is expensive due to redundant convolutions. Recent work [11, 28] has addressed this problem by performing the convolution stages on the full input image to efficiently create dense feature maps. These dense feature maps are then processed through convolution stages to replicate the fully-connected network at each pixel. An equivalent but efficient version of the sliding window model for a single resolution bank is shown in Fig 2. Note that due to pooling in the convolution stages, the output heat-map will be a lower resolution than the input image.

For our Part-Detector, we combine an efficient sliding window-based architecture with multi-resolution and overlapping receptive fields; the subsequent model is shown in Fig 3. Since the large context (low resolution) convolution bank requires a stride of $1/2$ pixels in the lower resolution image to produce the same dense output as the sliding window model, the bank must process four down-sampled images, each with a $1/2$ pixel offset, using shared weight convolutions. These four outputs, along with the high resolution convolutional features, are processed through a 9x9 convolution stage (with 512 output features) using the same weights as the first fully connected stage (Fig 1) and then the outputs of the low resolution bank are added and interleaved with the output of high resolution bank.

To improve training time we simplify the above architecture by replacing the lower-resolution stage with a single convolution bank as shown in Fig 4 and then upscale the resulting feature map. In our practical implementation we use 3 resolution banks. Note that the simplified architecture is no longer equivalent to the original sliding-window network of Fig 1 since the lower resolution convolution features are effectively decimated and replicated leading into the fully-connected stage, however we have found empirically that the performance loss is minimal.

Supervised training of the network is performed using batched Stochastic Gradient Descent (SGD) with Nesterov Momentum. We use a Mean Squared Error (MSE) criterion to minimize the distance between the predicted output and a target heat-map. The target is a 2D Gaussian with a small variance and mean centered at the ground-truth joint locations. At training time we also perform random perturbations of the input images (randomly flipping and scaling the images) to increase generalization performance.

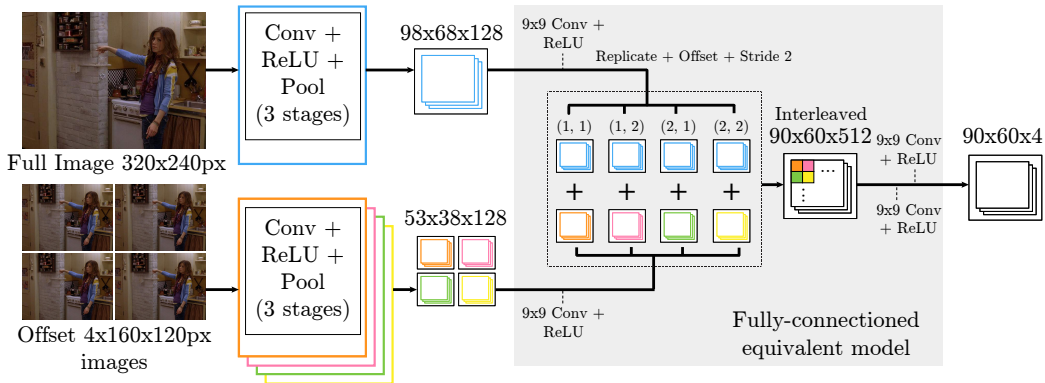

Figure 3: Efficient Sliding Window Model with Overlapping Receptive Fields

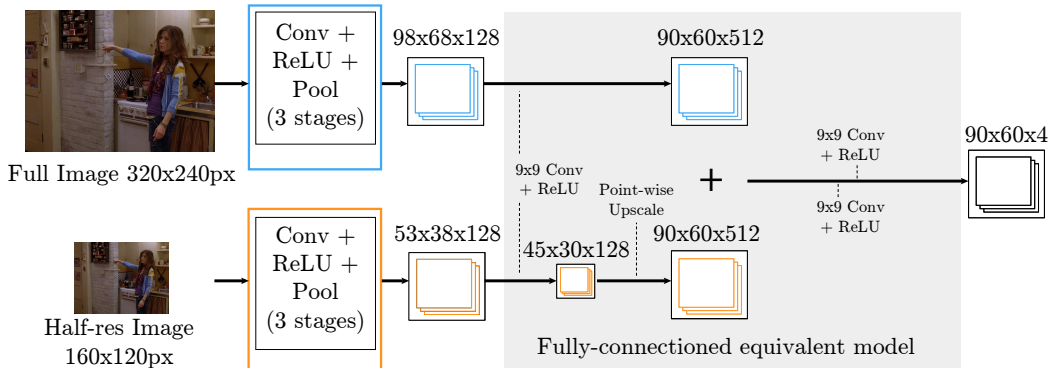

Figure 4: Approximation of Fig 3

## 3.2 Higher-Level *Spatial-Model*

The Part-Detector (Section 3.1) performance on our validation set predicts heat-maps that contain many false positives and poses that are anatomically incorrect; for instance when a peak for face detection is unusually far from a peak in the corresponding shoulder detection. Therefore, in spite of the improved Part-Detector context, the feed forward network still has difficulty learning an implicit model of the constraints of the body parts for the full range of body poses. We use a higher-level *Spatial-Model* to constrain joint inter-connectivity and enforce global pose consistency. The expectation of this stage is to not increase the performance of detections that are already close to the ground-truth pose, but to remove false positive outliers that are anatomically incorrect.

Similar to Jain et al. [15], we formulate the *Spatial-Model* as an MRF-like model over the distribution of spatial locations for each body part. However, the biggest drawback of their model is that the body part priors and the graph structure are explicitly hand crafted. On the other hand, we learn the prior model and implicitly the structure of the spatial model. Unlike [15], we start by connecting every body part to itself and to every other body part in a pair-wise fashion in the spatial model to create a fully connected graph. The Part-Detector (Section 3.1) provides the unary potentials for each body part location. The pair-wise potentials in the graph are computed using *convolutional priors*, which model the conditional distribution of the location of one body part to another. For instance, given that body part $B$ is located at the center pixel, the convolution prior $P_{A|B}(i, j)$ is the likelihood of the body part $A$ occurring in pixel location $(i, j)$. For a body part $A$, we calculate the final marginal likelihood $\bar{p}_A$ as:

$$\bar{p}_A = \frac{1}{Z} \prod_{v \in V} \left( p_{A|v} * p_v + b_{v \to A} \right) \tag{1}$$

where $v$ is the joint location, $p_{A|v}$ is the conditional prior described above, $b_{v \to a}$ is a bias term used to describe the background probability for the message from joint $v$ to $A$, and $Z$ is the partition

function. Evaluation of Eq 1 is analogous to a single round of sum-product belief propagation. Convergence to a global optimum is not guaranteed given that our spatial model is not tree structured. However, as it can been seen in our results (Fig 8b), the inferred solution is sufficiently accurate for all poses in our datasets. The learned pair-wise distributions are purely uniform when any pairwise edge should to be removed from the graph structure. Fig 5 shows a practical example of how the Spatial-Model is able to remove an anatomically incorrect strong outlier from the face heat-map by incorporating the presence of a strong shoulder detection. For simplicity, only the shoulder and face joints are shown, however, this example can be extended to incorporate all body part pairs. If the shoulder heat-map shown in Fig 5 had an incorrect false-negative (i.e. no detection at the correct shoulder location), the addition of the background bias $b_{v \to A}$ would prevent the output heat-map from having no maxima in the detected face region.

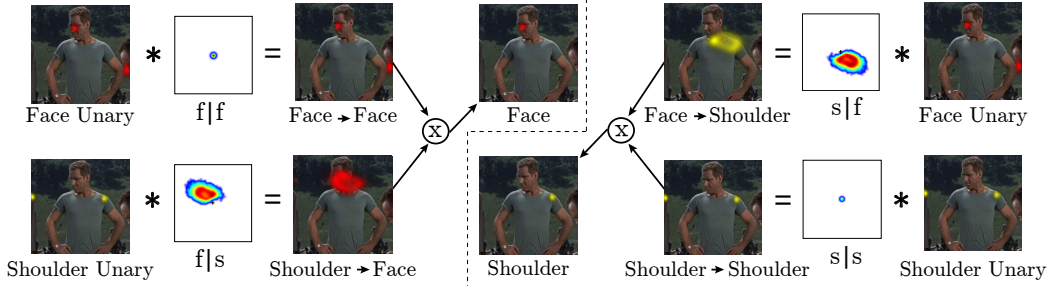

Figure 5: Didactic Example of Message Passing Between the Face and Shoulder Joints

Fig 5 contains the conditional distributions for face and shoulder parts learned on the FLIC [27] dataset. For any part $A$ the distribution $P_{A|A}$ is the identity map, and so the message passed from any joint to itself is its unary distribution. Since the FLIC dataset is biased towards front-facing poses where the right shoulder is directly to the lower right of the face, the model learns the correct spatial distribution between these body parts and has high probability in the spatial locations describing the likely displacement between the shoulder and face. For datasets that cover a larger range of the possible poses (for instance the LSP [17] dataset), we would expect these distributions to be less tightly constrained, and therefore this simple Spatial-Model will be less effective.

For our practical implementation we treat the distributions above as energies to avoid the evaluation of $Z$. There are 3 reasons why we do not include the partition function. Firstly, we are only concerned with the maximum output value of our network, and so we only need the output energy to be proportional to the normalized distribution. Secondly, since both the part detector and spatial model parameters contain only shared weight (convolutional) parameters that are equal across pixel positions, evaluation of the partition function during back-propagation will only add a scalar constant to the gradient weight, which would be equivalent to applying a per-batch learning-rate modifier. Lastly, since the number of parts is not known a priori (since there can be unlabeled people in the image), and since the distributions $p_v$ describe the part location of a single person, we cannot normalize the Part-Model output. Our final model is a modification to Eq 1:

$$\bar{e}_A = \exp \left( \sum_{v \in V} \left[ \log \left( \text{SoftPlus} \left( e_{A|v} \right) * \text{ReLU} \left( e_v \right) + \text{SoftPlus} \left( b_{v \to A} \right) \right) \right] \right) \qquad (2)$$

$$\text{where: } \text{SoftPlus} \left( x \right) = \frac{1}{\beta} \log \left( 1 + \exp \left( \beta x \right) \right), \frac{1}{2} \leq \beta \leq 2$$
$$\text{ReLU} \left( x \right) = \max \left( x, \epsilon \right), 0 < \epsilon \leq 0.01$$

Note that the above formulation is no longer exactly equivalent to an MRF, but still satisfactorily encodes the spatial constraints of Eq 1. The network-based implementation of Eq 2 is shown in Fig 6. Eq 2 replaces the outer multiplication of Eq 1 with a log space addition to improve numerical stability and to prevent coupling of the convolution output gradients (the addition in log space means that the partial derivative of the loss function with respect to the convolution output is not dependent on the output of any other stages). The inclusion of the *SoftPlus* and *ReLU* stages on the weights, biases and input heat-map maintains a strictly greater than zero convolution output, which prevents numerical issues for the values leading into the *Log* stage. Finally, a *SoftPlus* stage is used to

maintain continuous and non-zero weight and bias gradients during training. With this modified formulation, Eq 2 is trained using back-propagation and SGD.

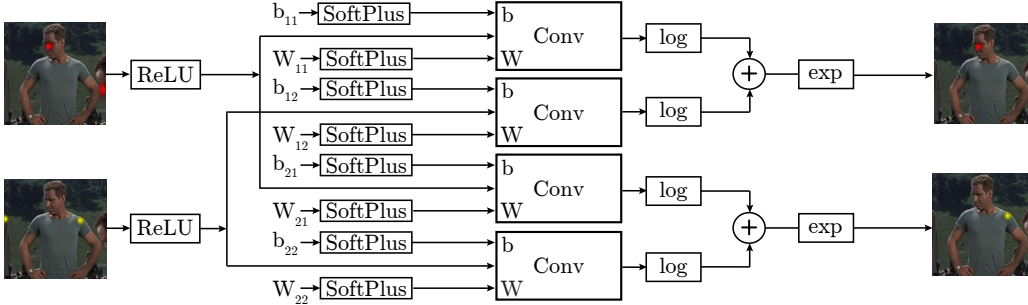

Figure 6: Single Round Message Passing Network

The convolution sizes are adjusted so that the largest joint displacement is covered within the convolution window. For our 90x60 pixel heat-map output, this results in large 128x128 convolution kernels to account for a joint displacement radius of 64 pixels (note that padding is added on the heat-map input to prevent pixel loss). Therefore for such large kernels we use FFT convolutions based on the GPU implementation by Mathieu et al. [19].

The convolution weights are initialized using the empirical histogram of joint displacements created from the training examples. This initialization improves learned performance, decreases training time and improves optimization stability. During training we randomly flip and scale the heat-map inputs to improve generalization performance.

### 3.3 Unified Model

Since our Spatial-Model (Section 3.2) is trained using back-propagation, we can combine our Part-Detector and Spatial-Model stages in a single *Unified Model*. To do so, we first train the Part-Detector separately and store the heat-map outputs. We then use these heat-maps to train a Spatial-Model. Finally, we combine the trained Part-Detector and Spatial-Models and back-propagate through the entire network.

This unified fine-tuning further improves performance. We hypothesize that because the Spatial-Model is able to effectively reduce the output dimension of possible heat-map activations, the Part-Detector can use available learning capacity to better localize the precise target activation.

## 4 Results

The models from Sections 3.1 and 3.2 were implemented within the Torch7 [6] framework (with custom GPU implementations for the non-standard stages above). Training the Part-Detector takes approximately 48 hours, the Spatial-Model 12 hours, and forward-propagation for a single image through both networks takes 51ms [1].

We evaluated our architecture on the FLIC [27] and extended-LSP [17] datasets. These datasets consist of still RGB images with 2D ground-truth joint information generated using Amazon Mechanical Turk. The FLIC dataset is comprised of 5003 images from Hollywood movies with actors in predominantly front-facing standing up poses (with 1016 images used for testing), while the extended-LSP dataset contains a wider variety of poses of athletes playing sport (10442 training and 1000 test images). The FLIC dataset contains many frames with more than a single person, while the joint locations from only one person in the scene are labeled. Therefore an approximate torso bounding box is provided for the single labeled person in the scene. We incorporate this data by including an extra "torso-joint heat-map" to the input of the Spatial-Model so that it can learn to select the correct feature activations in a cluttered scene.

The FLIC-full dataset contains 20928 training images, however many of these training set images contain samples from the 1016 test set scenes and so would allow unfair over-training on the FLIC test set. Therefore, we propose a new dataset - called *FLIC-plus* (http://cims.nyu.edu/~tompson/flic_plus.htm) - which is a 17380 image subset from the FLIC-plus dataset. To create this dataset, we produced unique scene labels for both the FLIC test set and FLIC-plus training sets using Amazon Mechanical Turk. We then removed all images from the FLIC-plus training set that shared a scene with the test set. Since 253 of the sample images from the original 3987 FLIC training set came from the same scene as a test set sample (and were therefore removed by the above procedure), we added these images back so that the FLIC-plus training set is a superset of the original FLIC training set. Using this procedure we can guarantee that the additional samples in FLIC-plus are sufficiently independent to the FLIC test set samples.

For evaluation of the test-set performance we use the measure suggested by Sapp et. al. [27]. For a given normalized pixel radius (normalized by the torso height of each sample) we count the number of images in the test-set for which the distance of the predicted UV joint location to the ground-truth location falls within the given radius.

Fig 7a and 7b show our model's performance on the the FLIC test-set for the elbow and wrist joints respectively and trained using both the FLIC and FLIC-plus training sets. Performance on the LSP dataset is shown in Fig 7c and 8a. For LSP evaluation we use person-centric (or non-observer-centric) coordinates for fair comparison with prior work [30, 8]. Our model outperforms existing state-of-the-art techniques on both of these challenging datasets with a considerable margin.

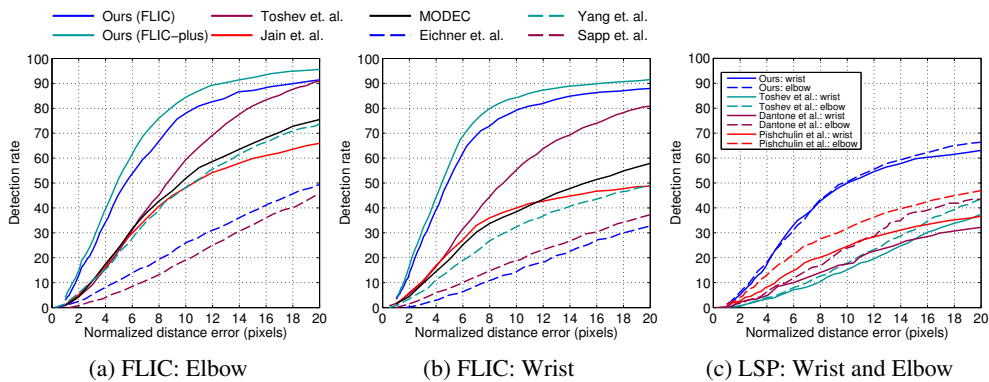

(a) FLIC: Elbow      (b) FLIC: Wrist      (c) LSP: Wrist and Elbow

Figure 7: Model Performance

Fig 8b illustrates the performance improvement from our simple Spatial-Model. As expected the Spatial-Model has little impact on accuracy for low radii threshold, however, for large radii it increases performance by 8 to 12%. Unified training of both models (after independent pre-training) adds an additional 4-5% detection rate for large radii thresholds.

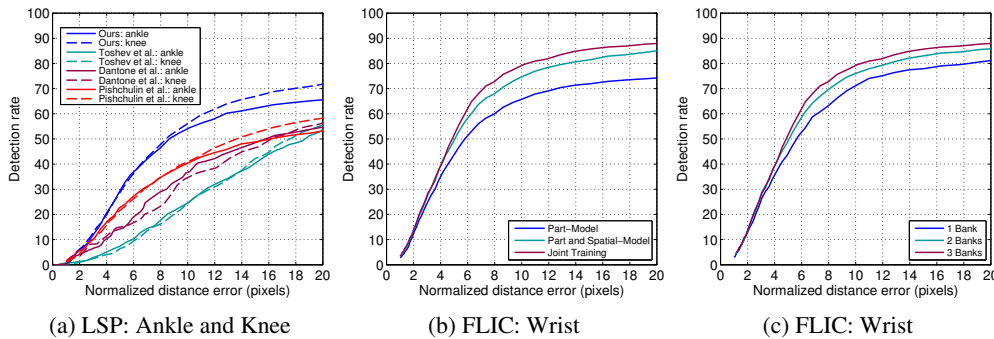

(a) LSP: Ankle and Knee      (b) FLIC: Wrist      (c) FLIC: Wrist

Figure 8: (a) Model Performance (b) With and Without Spatial-Model (c) Part-Detector Performance Vs Number of Resolution Banks (FLIC subset)

The impact of the number of resolution banks is shown in Fig 8c). As expected, we see a big improvement when multiple resolution banks are added. Also note that the size of the receptive fields as well as the number and size of the pooling stages in the network also have a large impact on the performance. We tune the network hyper-parameters using coarse meta-optimization to obtain maximal validation set performance within our computational budget (less than 100ms per forward-propagation).

Fig 9 shows the predicted joint locations for a variety of inputs in the FLIC and LSP test-sets. Our network produces convincing results on the FLIC dataset (with low joint position error), however, because our simple Spatial-Model is less effective for a number of the highly articulated poses in the LSP dataset, our detector results in incorrect joint predictions for some images. We believe that increasing the size of the training set will improve performance for these difficult cases.

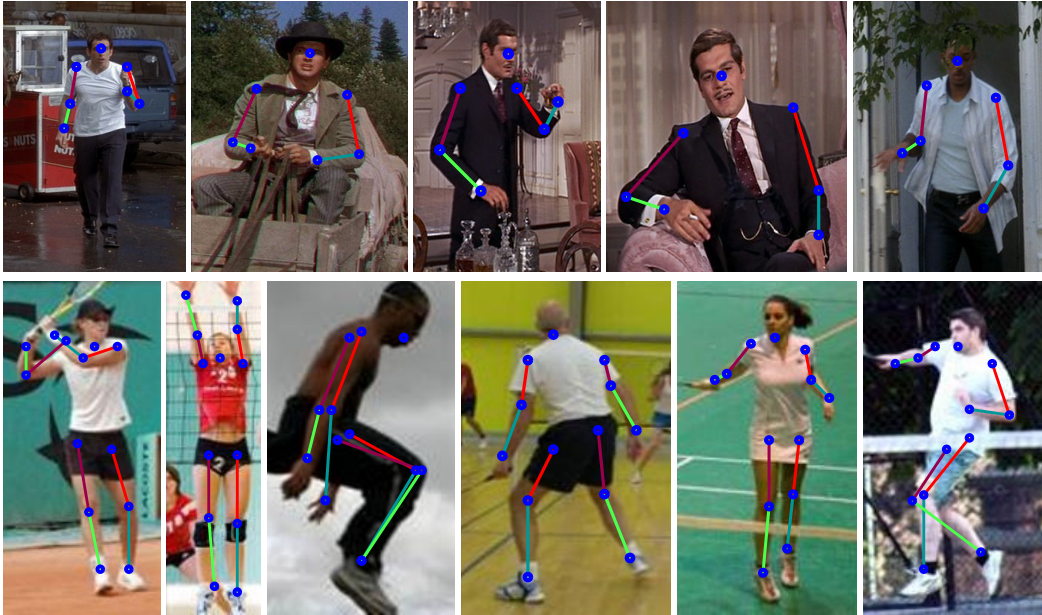

Figure 9: Predicted Joint Positions, Top Row: FLIC Test-Set, Bottom Row: LSP Test-Set

## 5   Conclusion

We have shown that the unification of a novel ConvNet Part-Detector and an MRF inspired Spatial-Model into a single learning framework significantly outperforms existing architectures on the task of human body pose recognition. Training and inference of our architecture uses commodity level hardware and runs at close to real-time frame rates, making this technique tractable for a wide variety of application areas.

For future work we expect to further improve upon these results by increasing the complexity and expressiveness of our simple spatial model (especially for unconstrained datasets like LSP).

## 6   Acknowledgments

The authors would like to thank Mykhaylo Andriluka for his support. This research was funded in part by the Office of Naval Research ONR Award N000141210327.

## Footnotes

[1]We use a 12 CPU workstation with an NVIDIA Titan GPU

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
