[Reviews · NeurIPS 2014]

Submitted by Assigned_Reviewer_13

This paper proposes a novel approach to human pose estimation, consisting of a deep convolutional network for part detection and a higher-level spatial model that is motivated as a graphical model, but actually incorporated into the overall deep network as a particular sub-net that has the plausible interpretation of performing a single round of message passing.

The system is trained in three steps. In the first two steps, the deep convolutional part detector and the spatial model are trained individually (the spatial message passing network uses the heat map output of the part detector), while in the third step, the unified network is jointly trained via back propagation.

Even though the convolutional part detector alone is already a state-of-the-art system, the spatial model is shown to improve results considerably, with even further improvements gained via the joint training procedure.

Quality:
The proposed approach is technically sound and the effectiveness of the proposed model architecture is demonstrated very convincingly through its strong empirical performance. Perhaps with the exception of the interesting spatial message passing network, the paper does not make many theoretical contributions, nor does it claim to.

Clarity:
The manuscript is exceptionally well-written and the quality of the figures and illustrations is outstanding. It is clear that an enormous amount of work went into preparation of the manuscript (and the whole project, for that matter).

Originality:
The convolutional part detector is based on a recent line of similar work exposing state-of-the-art performance and adds further practical tweaks that are perhaps better classified as brilliant engineering, rather than scientific advances. The spatial message passing network is novel to my knowledge, at least in this context. It is similar in nature to recent work on "inference machines" [R1], which should be cited.

Significance:
The experimental results presented in the paper are extremely strong, improving considerably on recent state-of-the-art work [36] from CVPR2014. It is very likely that the proposed model architecture will be adopted as a new baseline by many researchers and practitioners. As such, the proposed approach is likely to have significant impact.

[1] Ross, Stephane, et al. "Learning message-passing inference machines for structured prediction." Computer Vision and Pattern Recognition (CVPR), 2011 IEEE Conference on. IEEE, 2011.

Summary: The present manuscript is exceptionally well-written and very carefully prepared. It combines excellent engineering practices with novel modeling choices to obtain a high-quality pose estimation system that is clearly ahead of its competition. The approach is very likely to receive considerable attention by the computer vision community. For these reasons, I strongly recommend that the paper be accepted into the technical program of NIPS.

Submitted by Assigned_Reviewer_27

This work deals with the problem of human pose estimation in a single RGB image. A hybrid model is proposed, which combines the directed Convolutional Neural Networks (CNN) and an undirected graphical model (MRF). CNNs are used as body parts detectors, which produce a pixel-wise likelihood for the key joint locations. The undirected graphical model (MRF-like) is used to impose the spatial constrains among parts. In the pre-training stage, the CNNs and the undirected graphical model are trained separately. Then, the hybrid model is trained jointly using standard back-propagation and stochastic gradient descent.

The main novelty of this paper is the joint training of CNN and MRF for human pose estimation.

There are several major issues about this paper.

First, in the parameter learning part, the partition function in Equation 1 is directly dropped out from the final formulation without any justification. The partition function is a function of the parameters as well. It is not right to ignore it and just minimize the energy part.

Also, the joint model training is not clear from the paper. Only Section 3.3 discusses a little of the training procedure. The idea of combining CNN with MRF is interesting, but details on how to combine the two is missing. For the unified model, the learning of its parameters of two parts should be done simultaneously or at least the learning of the parameters of one part should affect those of the other part. Combining the MRF and CNN is not a trivial task and more details are needed.

How is the back-propagation and stochastic gradient descent applied to Equation 2 for joint training? In Equation 2, is the part A treated as the root part? Or, there are formulas for every body parts?

It is unclear how their model can produce joint angle estimates since their model is used to estimate body parts.

Another concern is about the experiments. Only the detection accuracy for a subset of body parts is shown in the paper (2 parts for each database), while there are more joint locations in the database. Why not give the results for all the joints?

There are also some minor issues. First, it’s not clear how to generate the pixel-level likelihood map using the CNN, since the output of the CNN has different size as the input image. Second, it’s not clear how the undirected graphical model is connected. Is it fully connected?
Summary: It’s novel that the CNN and the undirected graphical model are combined together for human pose estimation. However, it’s not clear how the hybrid model is jointly trained. There also are some issues with the experiments.

The authors should clarify that the spatial model that they used is NOT a MRF and it is only an extra layer on top of the part model to impose body pose prior. Doing so they can justify to drop the partition term. The authors should also study how their model, in particular the spatial model, generalize to different databases. Finally, it remains important to study the joint learning of the part model and the prior model instead of learning them separately.

Submitted by Assigned_Reviewer_31

TECHNICAL CONTENT & CLARITY:
The "deep convolutional network" part-detectors of Sec. 3.1 seem like a sensible application of recently developed deep learning methods. There does not appear to be significant novelty, but the architecture seems carefully engineered and is clearly described. One comment, if the "approximate" architecture in Figure 4 is used for the actual results, why bother to show the more complex architecture in Figure 3? There are no conceptual principles presented as to why one should be better than the other; the validation is experimental, so why not focus your explanatory effort on the thing you actually experiment with?

In Sec. 3.2, there is no valid joint distribution defined on the body part locations, and no non-trivial graphical structure; thus it seems inappropriate to call this a "graphical model". Instead there is a single round of message passing between all part pairs, as illustrated in Figure 5. This message passing has a form motivated by the sum-product BP algorithm, but need not correspond to BP (for example in a valid graphical model there would be constraints relating f|s and s|f which are not enforced here), and is not iterated to any mutually-consistent fixed point. A comment after Equation (2) says this formulation is not "exactly equivalent to an MRF", but I would argue more strongly, and say that the final formulation is rather distant from an MRF. The claims in earlier parts of the paper on integration with graphical models should be moderated to reflect this.

For related work which is more similar to this approach, see the "cascaded classification models" of Heitz 2008, or more distantly the "sum-product networks" developed by Domingos and colleagues. The approach is rather different from work like Bergtholdt 2010, which carefully treats the issues of inference and learning in fully-connected part-based graphs.

EXPERIMENTS:
Results on two contemporary pose estimation datasets are very good, improving on several strong baselines. A careful internal comparison also shows relative benefits of adding the final spatial message-passing layer, and doing a final round of back-propagation training to refine the full set of model parameters. These results are certainly the greatest strength of the paper.

I would have liked to also see an evaluation on the "Buffy the Vampire Slayer" dataset, which has been very widely studied in the pose estimation literature, and would this provide a wider set of baselines.

REFERENCES:
G. Heitz, S. Gould, A. Saxena, and D. Koller (2008). "Cascaded Classification Models: Combining Models for Holistic Scene Understanding." Advances in Neural Information Processing Systems (NIPS 2008).

Bergtholdt, Kappes, Schmidt, Schnorr, A Study of Parts-Based Object Class Detection Using Complete Graphs. International Journal of Computer Vision, Volume 87 Issue 1-2, March 2010, Pages 93 - 117.
Summary: For 2D human pose estimation problems, proposes to augment a deep convolutional-net body-part-detection architecture with a final layer modeling spatial interactions among parts. The approach is clearly described and seems well engineered, as there are strong performance improvements on a pair of reasonable benchmarks. However, the integration with graphical models is not as (conceptually or technically) strong as the abstract and introduction suggest.
Author Feedback
Author rebuttal: Reviewer 13:
Thank you for your positive feedback. We were not aware of the work by Ross et al., and we will cite it in the final manuscript.

Reviewer 27:
For joint training we employ a greedy (layer-wise) approach: we train the part detector individually, store the predicted heat-maps, then use them to train a spatial model. Finally, we train both together:

1. We FPROP through both model sections to generate a predicted output.
2. We BPROP through our error criterion to generate a error gradient w.r.t the SM output.
3. We BPROP through the SM to calculate the error gradient w.r.t the SM input.
4. Finally, we BPROP through the part-detector model.
5. With all gradients and output values defined we can now perform our SGD update for the “unified model”.

Note that we make no claim that this greedy approach is optimal, only that the above procedure improves our model performance. We have not thoroughly investigated training both models jointly from random initializations, but our current findings are that it results in worse performance. We will better clarify this procedure in the final manuscript

“the partition function in Eq 1 is directly dropped out from the final formulation without any justification”

We agree that we did not clearly justify this decision and we will fix this for the final manuscript.

There are 2 reasons why we do not include the partition function. Firstly, we are only concerned with the maximum output value of our network, and so we only need the output energy to be proportional to the normalized distribution. In addition, evaluation of the partition function is not necessary since both the part detector and spatial model parameters contain only shared weight (convolutional) parameters that are equal across pixel positions. Therefore, evaluation of the partition function during backpropagation will only add a scalar constant to the gradient weight, which would be equivalent to applying a per-batch learning-rate modifier. So it is indeed true that the magnitude of the gradient may differ (so both models are not equivalent), but because the search-direction is the same, in the limit our learning rate goes to zero, training the model with the partition function would converge to the same model as our proposed model and so it is removed for simplicity and to improve training time.

“How is the back-propagation and sgd applied to Equation 2 for joint training?”

For Eq 2 we can calculate the output gradient w.r.t the input and similarly calculate the output gradient w.r.t the model coefficients in closed form and BPROP through the layer as we would with any other network stage.

“In Eq 2, is the part A treated as the root part?”

From the paper: “we start by connecting every body part to itself and to every other body part in a pair-wise fashion in the spatial model.” This means that we do not have a tree structured graph, rather we use a pair-wise connected graph (so there is no root). For the final version we will add a figure showing graph connectivity and better clarify this in the text.

“It is unclear how their model can produce joint angle estimates since their model is used to estimate body parts.”

We am not sure we correctly understand your concern. We do not evaluate joint angles. The goal of our algorithm is to infer the 2D location of the body joints.

“Why not give the results for all the joints?”

As it is standard in the literature, we only show the performance for the wrist and elbow joints. These are significantly more challenging than other body parts. We did not include the performance of the other joints to save space and since we have no means to compare our results with other models. However, we are happy to add this data in a supplementary document.

“it’s not clear how to generate the pixel-level likelihood map using the CNN”

The convolutional network contains pooling layers, so the output resolution does not match the input resolution. To generate a per-pixel likelihood map in the input resolution we perform an upscaling from the heat-map resolution. For this we use nearest-neighbour upscaling. In general, one could use more precise upscaling, however we have found that it has only a minor impact on overall model performance.

Reviewer 31:
“if the "approximate" architecture in Fig 4 is used for the actual results, why bother to show the more complex architecture in Fig 3?”

We present the full architecture only to motivate our final design decisions for the approximate architecture. We will make this section more concise and focus on describing the final model used to avoid confusion.

“there is no valid joint distribution defined on the body part locations, and no non-trivial graphical structure; thus it seems inappropriate to call this a "graphical model””

We certainly understand that within many machine learning communities a graphical model is commonly defined as a *probabilistic* graphical model. However, we believe that - like a “factor graph” - our model is indeed a graphical model even though we do not explicitly deal with probabilities. Note that we use SGD to learn the “non-trivial” structure. While we include connections between all pairs in the graph, after training only a sparse number of joint connections contribute to the output distribution (for instance the model correctly learns only weak correlation between the face and foot joints). We understand that as written, our language might be too strong in connecting sum-product BP with our final formulation and so we will certainly tone down this language and make it clear that our model is inspired by (rather than equivalent to) BP in the earlier sections of the paper.

We will certainly add the citations that you mentioned to the final paper (Heitz & Bergotholdt).

We agree that including performance on the Buffy dataset would help relate our work to earlier literature and we will include this in the final manuscript.